# Ceramic Scaffolds in a Vacuum Suction Handle for Intraoperative Stromal Cell Enrichment

**DOI:** 10.3390/ijms21176393

**Published:** 2020-09-02

**Authors:** André Busch, Monika Herten, Marcel Haversath, Christel Kaiser, Sven Brandau, Marcus Jäger

**Affiliations:** 1Department of Orthopedics, Trauma and Reconstructive Surgery, St. Marien Hospital Mülheim an der Ruhr, D-45468 Mülheim/Ruhr, Germany; An.Busch@contilia.de; 2Department of Orthopedics and Trauma Surgery, University Hospital Essen, University of Duisburg-Essen, D-45147 Essen, Germany; Christel.Kaiser@uk-essen.de; 3Department of Orthopedics, St. Vinzenz Hospital Düsseldorf, 40477 Düsseldorf, Germany; M.Haversath@gmx.de; 4Department of Otorhinolaryngology, University Hospital Essen, University of Duisburg-Essen, D-45147 Essen, Germany; Sven.Brandau@uk-essen.de; 5Chair of Orthopedics and Trauma Surgery, University of Duisburg Essen, D-45147 Essen, Germany

**Keywords:** bone defect, bone remodeling, surgical suction, ceramic filter, *β*-TCP, allograft, total joint arthroplasty

## Abstract

During total joint replacement, high concentrations of mesenchymal stromal cells (MSCs) are released at the implantation site. They can be found in cell–tissue composites (CTC) that are regularly removed by surgical suction. A surgical vacuum suction handle was filled with bone substitute granules, acting as a filter allowing us to harvest CTC. The purpose of this study was to investigate the osteopromotive potential of CTC trapped in the bone substitute filter material during surgical suction. In the course of 10 elective total hip and knee replacement surgeries, *β*-tricalcium-phosphate (TCP) and cancellous allograft (Allo) were enriched with CTC by vacuum suction. Mononuclear cells (MNC) were isolated from the CTC and investigated towards cell proliferation and colony forming unit (CFU) formation. Furthermore, MSC surface markers, trilineage differentiation potential and the presence of defined cytokines were examined. Comparable amounts of MNC and CFUs were detected in both CTCs and characterized as MSC‰ of MNC with 9.8 ± 10.7‰ for the TCP and 12.8 ± 10.2‰ for the Allo (*p* = 0.550). CTCs in both filter materials contain cytokines for stimulation of cell proliferation and differentiation (EGF, PDGF-AA, angiogenin, osteopontin). CTC trapped in synthetic (TCP) and natural (Allo) bone substitute filters during surgical suction in the course of a joint replacement procedure include relevant numbers of MSCs and cytokines qualified for bone regeneration.

## 1. Introduction

Bone and its marrow is a highly vital tissue with multiple functions including mechanical stability, hemostasis, immune response and tissue regeneration [1]. It is the only tissue which can heal without fibrous scar after damage and is able to adapt to biomechanical forces (“bone remodeling”) [2]. However, in bone defects of critical size, the self-healing capacity of the bone tissue is limited, resulting in delayed union or non-union [3]. The manifold reasons for impaired bone healing are often influenced by individual factors such as local vascular supply, infection status, comorbidities and others [4]. Besides sufficient stabilization of the bone defect by internal or external fixations (osteosynthesis), the gold standard treatment for non-unions is autologous bone grafting [5]. However, the amount of autologous bone is limited and the procedure is associated with further comorbidity.

Therefore, a large amount of both natural (allo-/xenografts) and synthetic (e.g., tricalcium phosphate (TCP), hydroxyapatite (HA), calcium sulfate, calcium carbonate, composites) bone substitutes have been applied in orthopedic and trauma surgery for a long time [6,7,8,9]. In the last few years, mesoporous bioactive glasses (MBGs) are becoming increasingly popular as bone substitute materials [10]. In contrast to autologous bone, these materials are not vital and mainly act as an osteoconductive scaffold for local tissue ingrowth. Other bone substitutes pledge additional osteoinductive potential, with favorable effects on proliferation and differentiation of local cells. Osteoinductive materials are potent to induce new bone formation even in a non-osseous environment by recruiting undifferentiated and pluripotent progenitor cells, followed by differentiation into bone-forming cells. Bone morphogenetic protein (BMP) and platelet-derived growth factor (PDGF) coated biomaterials are typical candidates for these osteoinductive materials [11]. As a consequence, high amounts of local osteoprogenitors at the bone defect site are preferable.

Recently, multiple studies present promising results regarding the use of MSCs and various growth factors in the treatment of fractures and non-unions, as well as synergistic effects observed when combined [12]. Additionally, clinical trials suggested that the application of MSC and scaffolds can enhance osteogenesis in bone defects [13,14,15]. However, the harvesting of these cells intraoperatively is time-consuming, there are sterility concerns as the aspirate may leave the sterile field, and there is also a risk of changing the biology of the cells depending on the following processing of the tissue [16,17,18,19,20].

Apart from the above, osteopromotive scaffolds promote bone regeneration within an osseous tissue and enhance bone formation during various stages of bone healing. This property can be achieved by additional substances such as local tissue, platelet-rich plasma or coatings (e.g., collagen, fibrin) [21].

In orthopedics and trauma surgery, hard tissue opening or damage is frequent and associated with a relevant amount of bone marrow leaking out of the bone. Furthermore, the surgical approach and/or trauma includes also soft tissue damage and hematoma. As a consequence, not only bone fragments but also small pieces of muscle, fat, connective tissue and vessels are liberated from the surrounding tissue. Some of these are removed by surgical suction; some are left on the site. Moreover, the injured and crushed tissue releases soluble components such as loco-typical cells including progenitors, as well as intra- and extracellular fluid (proteins, lipids, water and electrolytes), cytokines, growth factors and others [22]. If internal fixation is carried out, parts of these components adhere to the implant’s surface and act as preconditioning protein film at the bone–implant interface, promoting osseointegration (“implant proteome”) [23]. At the very least, this environment induces aseptic inflammation and provides the basis for later bone and tissue healing.

During surgery, most surgeons use vacuum suction devices to obtain an optimal view of the surgical site. Moreover, these systems can be combined with cell saver devices if relevant blood loss is assumed (autologous blood transfusion). Especially during total joint replacement or other types of operations demanding a more extensive surgical approach, relevant amounts of the abovementioned mixture of cytokines, growth factors and cells and tissue (cell–tissue composites, CTCs) will become lost by vacuum suction [24,25]. In a previous study, we showed that surgical vacuum filters were able to concentrate tissue with relevant amounts of MSCs, presenting a new potent source of autologous regeneration material [22].

To make this tissue useful for a possible clinical application in the future, a granule-based bone substitute filter was evaluated in the present study towards the retention of progenitor cells and cytokines qualified for bone regeneration.

To our knowledge, this is the first device which is able to enrich autologous tissue onto a bone substitute without any additional surgical approach or other time-consuming interoperative procedures. Moreover, the cell/tissue–TCP/allograft composite does not require further processing outside the surgical site.

There are many research groups who are working on solutions for bone regeneration onto biomaterials [26]. In contrast to other investigators, we intend to present a simple, time-saving and patient-friendly solution. Compared to other systems such the reamer-irrigator-aspirator [27] or bone aspiration techniques with or without density gradient cell centrifugation (e.g., the Harvest^®^ system [28], ASPIRE™ Bone Marrow Harvesting System [29]), no additional surgical approach or device material is required. Other groups prefer platelet-rich plasma (PRP) or other products from the peripheral blood to treat bone defects [30,31].

Although the content of platelet lysates is highly promising for regenerative approaches in translational orthopedics [32] and these systems are also non-invasive (i. v. punction), they leave the surgical site and do not penetrate porous bone substitutes as a vacuum system can. At the very least, most systems designed for bone regeneration are at a pre-clinical stage [33]. A simple solution, such as a vacuum suction handle combined with a bone substitute, might enter the market as a medicinal product much more easily by the extension of an established certificate (suction handle only).

## 2. Results

### 2.1. Comparison of the MNC Yield, Stemness Character and Proliferation Potential

The average weight of the samples harvested was comparable between the two bone substitute materials (BSMs) (*p* = 0.462), with 13.8 ± 5.6 g for allogenic BSM (Allo) range (7.0–22.8) and 15.9 ± 4.5 g for the synthetic BSM (TCP) range (12.4–26.4). In all 10 patients, mononuclear cells (MNCs) could be isolated and cultured from both BSM groups. The total amount of harvested mononuclear cells (MNCs) was comparable between the groups (*p* = 0.145), with 1.94 ± 1.32 × 10^10^ range (0.81–4.51 × 10^10^) for Allo and 1.26 ± 1.03 × 10^10^ range (0.31–3.44 × 10^10^) for TCP (Figure 1A). The order of the BSM used in the surgical vacuum suction device within the surgery did not influence the MNC yield (*p* = 0.825). The weight of the samples correlated with the amount of harvested MNCs, with r = 0.606 (Pearson) and with a significant *p*-value of 0.006.

A colony-forming unit (CFU) assay was performed with MNCs of each group (Allo vs. TCP). We found higher values in CFU for MNCs harvested from the TCP, with 1.076 ± 1.073 ×10^6^ MNCs and a range of (0.03–2.47 × 10^6^ MNC) in the surgical vacuum suction device compared to the Allo (0.82 ± 0.75 × 10^6^ MNC), with a range of (0.02–1.99 × 10^6^ MNC) (Figure 1B), but this difference was not significant (*p* = 0.581).

From the CFU assay results, the number of cells with the potential for forming a colony from 1 × 10^6^ MNC was determined and used for calculating the theoretical number of potential MSCs in the different BSM samples. Regarding the ratio of potential MSCs per MNC harvested, the values were slightly higher in Allo, with 12.8 ± 10.2‰, median of 12.1 and range of (0.3–30.9) compared to TCP, with 9.8 ± 10.7‰, median of 7.0 and a range of (0.7–28.7) (*p* = 0.550). Per filter handle, the average calculated number of MSCs was 12,846 ± 10.214, range (293–30,886) for Allo and 9830 ± 10,720, range (663–28,710) for TCP.

The proliferation potential determined as generation time was comparable for Allo and TCP in P1 and P2, with 8.2 ± 2.9 days, range (3.5–12.8) Allo vs. TCP 8.4 ± 1.1 days, range (5.1–14.1) for P1 (*p* = 0.604). In P2, the values were 6.9 ± 2.9 days, range (3.3–11.8) for Allo vs. TCP 6.3 ± 2.5 days, range (4.1–10.7), respectively (*p* = 0.998) (Figure 1C). It was slightly lower for the cells in P1 compared to P2 for both groups (*p* = 0.062).

### 2.2. Differentiation Potential

The occurrence of MSCs was controlled by the expression of typical markers via flow cytometry. Here, cells expressed no hematopoietic markers (CD31^−^, CD34^−^, CD45^−^) but all samples showed significant expression of the mesenchymal stroma cell markers (CD29^+^, CD73^+^, CD90^+^, CD105^+^) (Figure 2). There were no differences in the expression type between the two biomaterials.

No qualitative differences could be found in the lineage specific differentiation between the different BSM groups. Figure 3 demonstrates the characteristic cytochemical staining of calcium in the extracellular matrix components with alizarin red (osteoblasts), of the intracellular oil vacuoles with oil red (adipoblasts) or of the glycosaminoglycans with alcian blue (chondroblasts).

### 2.3. Cytokine Pattern in the Cell–Tissue Composite

The determination of the relative levels of 104 selected human cytokines, chemokines, growth factors, hormones and other soluble proteins in the cell–tissue composite from the surgical vacuum filter revealed no differences in their quality and in their semiquantitative rating (Figure 4). In both cell–tissue composites, the following proteins could be detected:

#### 2.3.1. Cytokines

Macrophage migration inhibitory factor (MIF) (#19), chemokine (C–C motif) ligand 5 (CCL5)/RANTES (#24), chemokine (C–X–C motif) ligand 4 (CXCL4)/PF4 (#23) and C–X–C motif chemokine 5 (CXCL5)/ENA-78 (#13).

#### 2.3.2. Growth Factors

Epidermal growth factor (EGF) (#11) and platelet derived growth factor (PDGF-AA) (#22).

#### 2.3.3. Hormones

Adiponectin (Acrp-30) (#1), resistin (XCP1) (#26).

#### 2.3.4. Other Signaling Proteins

Vascular cell adhesion molecule 1 (VCAM-1)/CD106 (#34), platelet endothelial cell adhesion molecule (PECAM)/CD31 (#32), endoglin (CD105) (#14), urokinase receptor (uPAR)/CD87 (#30), osteopontin (BSP-1/BNSP) (#21), thrombospondin-1 (#29), apolipoprotein A-I (ApoA1) (#2) and angiogenin (#3).

#### 2.3.5. The Transport/Binding Proteins (BP)

Interleukin-18-binding protein (IL-18 BP) (#17), insulin-like growth factor-binding protein 2 (IGFBP-2) (#15), insulin-like growth factor-binding protein 3 (IGFBP-3) (#16), retinol-binding protein 4 (RBP4) (#25), sex hormone-binding globulin (SHBG) (#28) and vitamin D BP (#31).

#### 2.3.6. Proteins Pertained to the Immune System

Complement component C5/C5a (#4), complement factor D (CFD/adipsin) (#7), C-reactive protein (CRP) (#8), CD14 (#5), lipocalin-2 (NGAL) (#18) and hepatitis A virus cellular receptor 2 (HAVCR2)/TIM-3 (#33).

#### 2.3.7. Other Proteins

Basigin (extracellular matrix metalloproteinase inducer EMMPRIN/CD147) (#12), cystatin C (#9), matrix metalloproteinase-9 (MMP-9)/gelatinase B (#20), serpin E1 (nexin) (#27), dipeptidyl peptidase-4 (DPP4)/CD26 (#10) and chitinase3 like 1 (CHI3L1/YKL-40) (#6).

## 3. Discussion

This study clearly demonstrated that bone substitutes placed in a surgical vacuum filter device were able to collect tissue with relevant amounts of MSC. Comparing allograft and *β*-TCP, no significant differences could be found in the capability to enrich MSCs and growth factors. Furthermore, we found neither biomaterial-associated differences for the amount of MNC harvested nor variations in the stemness character of the cells (number of CFU). Cells cultivated from both bone substitutes could be characterized as MSCs without any differences in their (i) adherence to the plastic surface, (ii) typical surface marker expression profile and (iii) differentiation into the three lines, osteoblasts, adipoblasts and chondroblasts.

### 3.1. Influence of the MSC Number on Bone Regeneration

Some data in the literature suggest that the dose and volumetric concentration of autologously applied MSCs appears to be critical for in vivo bone regeneration, rather than their purity [15,34]. Regarding our experiments, it is finally not clear if the amount of MSC collected within the BSM equipped vacuum filter handle is sufficient to repair a bone defect. In the present study, the use of the surgical vacuum suction device during 10 min of surgery harvested an amount of 12,846 MSCs [293–30,996] in Allo and 9830 [663–28,710] for TCP within one vacuum suction handle application. Hernigou et al. evaluated the necessary amount of transplanted progenitor cells for the treatment of non-union and demonstrated a significant positive correlation between the volume of mineralized callus after four months and the number and concentration of fibroblast colony-forming units in the graft [15]. Bone union was obtained in 53 of 60 patients (88.3%) with bone marrow injected into the non-unions containing >1500 progenitors/cm^3^ and an average total of 54,962 ± 17,431 progenitors per defect. If significantly lower concentrations (634 ± 187 progenitors/cm^3^; total number 19,324 ± 6843 of progenitors) were injected into the non-union site, bone union was not obtained in 11.5% of the patients [35].

Similarly, Le Nail et al. treated 43 patients with open tibial fractures with a risk of developing non-unions or presenting non-unions with injections of bone marrow concentrate. They postulated a minimum threshold of 360 × 10^3^ transplanted progenitors amounting to successful healing [36].

Although these data are difficult to compare to the study of Hernigou, the application of the vacuum filter handle for 10 min during surgery could not harvest the same amount of progenitor cells as isolated by bone marrow aspiration [15]. However, the surgical vacuum suction device guarantes wetting of the bone substitute as well as cellular retaining without any additional invasive procedure and without further tissue or cell processing [17,37]. It is unclear whether a longer application of the vacuum is able to increase the amount of MNC in the filter biomaterial. Moreover, we cannot exclude the influence of the plastic material of the suction device.

### 3.2. Influence of the Plastic Container and Vacuum

In our study, the suction device was made of polystyrene (OP-Flex™ FilterFlow™, ConvaTec, Deeside, UK), which is not qualified for orthopedic in vivo application. During the suction procedure, it is unlikely but possible that polystyrene particles might detach from the surface of the suction device and come into contact with the ceramic bone substitute. Due to patient safety issues, the authors recommend a more biocompatible and inert material (e.g., polymethyl methacrylate, PMMA) if clinical application is intended. Following this idea, a biocompatible surgical suction handle has been developed in a project granted by the European Union (EU) (BoneFlo^®^, TissueFlow).

Besides the application site (including cellular components) [37,38,39], the type of vacuum (level, duration) [17,37], the suction handle’s material [38,39] and the type of bone substitute [40,41] might have an impact on in vitro results.

### 3.3. Impact of the Biomaterial and Regulatory Aspects

In orthopedics and trauma surgery, ceramic bone substitutes are frequently used due to mechanical (e.g., compression resistance) and biological properties (e.g., advanced microstructure with interconnecting pores mimicking autologous bone) [42,43,44]. However, we found no significant differences comparing cortico-spongious chips (allograft) with porous TCP granules in our study. This could be a hint that the initial contact between components of the CTC and the biomaterial are less relevant for following cell proliferation and differentiation than the influence of the surrounding tissue in vivo. The experiences of more than three decades in bone substitute (e.g., hydroxyapatite (HA), TCP) application in orthopedics support this theory: none of the commercial products have prevailed or dominated the market to date.

However, it is noted that autologous bone grafts are still considered the gold standard, presenting all necessary properties as osteoconductivity, osteoinductivity and osteogenesis [45]. However, the application of new coating strategies and the development of composites (e.g., polycaprolactone-tricalcium phosphate (PCL-TCP)) have the potential to improve currently available materials [46,47].

Another point of discussion is how to combine cells and BSM for bone regeneration. Here, the BSM surface and the role of MSCs seems to be a key factor: in bone tissue engineering, autologous MSCs are harvested from different sources (e.g., bone marrow, adipose tissue). One option is to cultivate MSCs ex vivo in order to obtain sufficient amounts of cells which can be re-transplanted together with BSM in a second step. Here, two approaches exist: (i) cells are either left undifferentiated and implanted together with the BSM (direct loading of the cells) or (ii) cells are pre-seeded onto the scaffold and differentiated towards osteogenic precursors in 3D culture of the respective BSM before implantation [17,37]. The temporary cultivation of MSCs requires considerable logistical effort to ensure the quality of the cell therapy treatment [48] and the economic reimbursement is not given or remains questionable. Within the European Union (EU), the ex vivo cultivation procedure and application of cells requires not only good manufacturing practice (GMP) standards but also a license of national authorities for the production of cellular products in accordance with advanced therapy medicinal products (ATMP).

These regulations restrict in vitro tissue engineering for bone defects. However, Léotot et al. (2015) demonstrated that direct cell loading into a scaffold during surgery is more efficient for bone regeneration compared to pre-seeded BMCs for 7 days in vitro on HA/β-TCP [49].

An alternative method for enrichment of bone substitute materials with autologous MSCs is the application of concentrated bone marrow with BSM into the bone defect in a single step procedure during surgery. Several clinical studies showed the efficacy and safety using bone marrow concentrate (BMC) [12,13,14,15]. Although various commercial bone marrow concentration systems are available, which mostly work with gradient centrifugation, these will hardly be a generally accepted treatment option, due to official requirements and relevant uncertainty among orthopedic surgeons [50]. In Germany, the use of BMC for the treatment of bone defects has been limited by a regulation of the national equivalent to the FDA, the Paul Ehrlich Institute (PEI), which declared the BMC a new therapeutic agent requiring regulatory approval in 2009 [51].

### 3.4. Conditioning Systems, Composites and Growth Factors

Methods other than centrifugation for the enrichment of cells include filtration using ceramic materials with a certain selectivity for MSCs [52]. Until now, no reliable conclusion was found to explain the greater retaining of MSCs to carrier materials compared to other nuclear cells. Chu et al. (2018) showed that filtration of bone marrow cells through various BSMs led to an enrichment of MSCs and their direct adhesion to the surfaces of porous BSM [53]. Importantly, the filtration enrichment method allows MSC concentration without interference from exogen agents, revealing no significant changes in phenotypic characteristics of the MSCs, osteogenetic fate, specific antigens, gene expression profile, cell cycle stage and apoptosis rate compared to before filtration [37,44]. However, in the abovementioned study, additional bone marrow was collected from the anterior superior iliac spine, posing the risk of donor-site morbidity. In a recent study, our research group could demonstrate that surgical vacuum filters collecting tissue composites during orthopedic surgery are able to concentrate tissue with relevant amounts of MSCs. Moreover, the CFU numbers from the material collected in vacuum filters harvested during arthroplasties were superior to those from bone marrow and cancellous bone [22].

Established methods to harvest MSCs, such as RIA (reamer-irrigator-aspirator) require additional technical devices and consume time during surgery [54]. With an advanced vacuum suction handle, no additional devices are needed after placing ceramic bone substitutes into a surgical vacuum filter.

In the present study, the protein array of both CTC groups revealed the presence of relevant cytokines and growth factors (GF) for cellular proliferation and differentiation such as EGF, PDGF-AA, angiogenin as well as osteopontin as important factors in bone remodeling.

The cocktail of documented cytokines and growth factors in vitro imitates the situation in vivo during bone healing, where vascularization is a relevant step: angiogenin is a potent growth factor which is able to stimulate new blood vessel formation [55]. Platelet-derived growth factor (PDGF) is released from platelets following tissue injury and enhances cell migration (chemotaxis) and proliferation (mitogenesis) [56].

Toosi et al. precisely described in a review a range of growth factors and cytokines provided by bone resident MSCs to support cell growth following injury and demonstrated how GF signaling plays a critical role in regulating osteogenesis, chondrogenesis and bone/mineral homeostasis [57]. It is known that the delivery of exogenous GFs, i.e., BMP-2, to the non-union bone fracture site remarkably improves healing results [58].

Nevertheless, the application of growth factors may have drawbacks. Using supraphysiological dosages of growth factors to compensate for the shorter duration of activity in the in vivo milieu increases the risk of adverse effects such as excessive soft tissue inflammation and the formation of excessive ectopic bone [59]. Concerning their concentrations, in a recent study, their concentration in freshly isolated adipose tissue was correlated to the proliferation and migration capacity of the MSCs derived from the respective sample [60]. The proliferation and migration capacity of MSCs strongly depended on the GF content of the tissue but in an inversely proportional manner: the lower the GF content of the tissue, the higher was the proliferation and migration capacity of the respective MSC population contained in the adipose tissue (AT) and vice versa. Furthermore, they found that supplementation with recombinant GFs led to a significant enhancement of proliferation and migration of the AT-resident MSCs only in samples with low but not with higher growth factor contents [60]. This inefficiency of GFs to enhance MSC proliferation and migration in samples with high GF content indicated a GF-mediated negative feedback mechanism leading to impaired GF signaling in MSCs obtained from these tissue samples [59]. Further studies are necessary to investigate the impact of the growth factor content on the proliferation of MSCs derived from the filter in vacuum suction handles.

Our study has some limitations. The definite source of the MSCs and growth factors in the filter device cannot be determined, since various tissue compartments (subcutaneous fat, connective tissue, muscle tissue, periosteum, vessels/blood, bone and bone marrow) were affected during surgery. We cannot differentiate between trapping of the cells among the biomaterial chips or cellular retaining to the bone replacement material. Furthermore, the suction procedure was performed in a clinical scenario and not in a laboratory or on an experimental scale. It was done individually to provide an optimal view of the surgical site. Moreover, the weight of the BSM after the vacuum suction varied for both BSMs. Particularly, in some cases, a few BSM granules were spilled within the filling process; also, the weight of the resulting bone substitute material with CTC material varied from patient to patient, especially in obese patients.

Another limitation is that the tissue samples were obtained from two different surgical procedures (total hip or knee replacement). Therefore, differences in the harvested cells and growth factors might be based on the place of origin. In a recent published study, it was demonstrated that MSCs from the femur possess significantly higher osteogenic potential compared to the acetabulum [61]. Since, in both procedures (hip and knee replacement surgery), bone marrow from the femur is gathered, the different surgical locations may not be crucial for different results. Concerning the amount of MNC harvested, there was no difference in the order of the material used. Additionally, the low number of samples does not allow conclusions about statistical differences.

By and large, we demonstrated that MSCs and relevant cytokines for bone healing can be isolated in vitro and harvested by a combined surgical vacuum–BSM filter system.

## 4. Materials and Methods 

### 4.1. Patients

Following a prospective design, the patient cohort consisted of 10 patients (5 females, 5 males, mean age of 66.8 ± 9.1 years) with advanced osteoarthritis who qualified for elective total hip (*n* = 6; four females, two males) or knee (*n* = 4; one female, three males) replacement. The study protocol was approved by the local ethical committee (19-8822-BO, 19-02-2020, Ethical Board of the Medical Faculty of the University Duisburg-Essen, Germany; available in the Appendix A). All individuals had given their written informed consent prior to surgery.

Exclusion criteria were malignant or infectious diseases, immuno-suppressive drugs and an age under 18 years.

### 4.2. Surgical Vacuum Suction Device Used during Surgery

During surgery, a conventional surgical vacuum suction handle (OP-Flex™ FilterFlow™, ConvaTec, Deeside, UK) was filled with bone substitute material (TCP or Allo) and used for 10 min. The surgical procedure during this timeframe comprised a Harding–Bauer approach including femoral neck osteotomy (total hip replacement) or a mediopatellar approach to the knee joint up to intramedullary reaming of the distal femur (total knee replacement).

After this, the suction handle was replaced by a new device with the respective other bone substitute material. The handle was then used for another 10 min. During this time, the acetabulum was reamed (total hip replacement). In knee patients, jig-guided femoral osteotomies were performed. The sequence of the bone substitute selection was randomized before each operation. The granules used were as follows:

Allograft bone substitute material: maxgraft^®^ ortho, spongiosa granulate (Cells and Tissue Bank Austria gGmbH, Krems, Austria); granule size 2–8 mm; granule volume 10 mL.Synthetic *β*-TCP [Ca_3_(PO_4_)_2_]: Cerasorb M^®^, (Curasan AG, Kleinostheim, Germany); granule size 5–8 mm; granule volume 10 mL.

The surgical suction handle including the bone substitute and retained autologous material after ten minutes of suctioning was stored under sterile conditions and transported to the laboratory for further in vitro analysis. Suction process was performed with negative pressure of 20 mmHg using a fixed vacuum pump (Draeger AG, Lübeck, Germany) and a connective tube made of polyvinyl chloride (PVC) (Extrude TM, Surgery ApS, Birkerod, Denmark) (Figure 5).

### 4.3. Isolation and Cultivation of the Cells

The sealing cap of the surgical vacuum sucker was opened under the laminar flow bench and the internal plastic filter was removed from the holding groove (Figure 5A,B). The ceramic granules and retained CTC were emptied into a petri dish and weighed (Figure 5C). The mixture of BSM and cell–tissue composite (Figure 5E,F) was incubated with streptokinase (3 KU; S3134, Sigma Aldrich, Steinheim, Germany) in 10 mL phosphate-buffered saline (PBS) at RT for 15 min. The liquid phase was removed and filtered through nylon mesh (pore size 70 µm, Falcon, Heidelberg, Germany). An aliquot of 1 mL was allocated for further human cytokine array analysis and stored at −80 °C. The filtered solution was further diluted with PBS and centrifuged at 300× *g* at RT for 10 min. The pellet was resuspended and washed in PBS and centrifuged as before. The resulting cell pellet was resuspended in PBS and the number of mononuclear cells (MNC) was counted. Then, 3 × 10^7^ MNCs of each BSM type per patient were allocated for CFU assay. The remaining MNCs were layered onto a Ficoll (Ficoll Paque™ Plus, density 1.078 g/mL, GE Healthcare, Freiburg, Germany) gradient in SepMate™ tubes (STEMCELL Technologies Inc, Vancouver, Canada) and centrifuged at 1200× *g* at RT for 15 min (full acceleration 9/9, decreased deceleration 7/9). The interphase was collected in PBS and centrifuged at 300× *g* at RT for 10 min. The cell pellet was resuspended in medium and cultured in T25 tissue flasks at 37 °C and 5% (*v/v*) CO_2_. The medium consisted of low-glucose Dulbecco’s Modified Eagle’s Medium (DMEM) supplemented with l-glutamine (GlutaMAX™, both Gibco, Life Technologies, Darmstadt, Germany), with 10% (*v/v*) fetal calf serum (FCS, Biochrome, Berlin, Germany), 100 U/mL penicillin (Sigma Aldrich, Taufkirchen, Germany), 0.1 mg/mL streptomycin (Sigma Aldrich) and 1 mM sodium pyruvate (Sigma Aldrich). Medium was changed every 3rd day. At 80–90% confluence, the adherent cells were detached by accutase (600 U/mL, Gibco/Life Technologies, Carlsbad, CA, USA), counted and seeded at a density of 6.5 ± 2.5 × 10^3^ cells per cm^2^ in flasks. The cellular doubling time was determined and defined as generation time. Cells of 3rd passage were used for flow cytometry analysis and cell differentiation testing.

### 4.4. Colony Forming Unit (CFU) Assay

MNCs of each BSM type were cultivated in 6-well plates with 1 × 10^6^, 4 × 10^6^ and 10 × 10^6^ cells per well as duplicates at a final cell density of 1 × 10^5^, 4 × 10^5^ and 10 × 10^5^ MNC/cm^2^. Medium was changed every 3rd day. At day 14, cells were washed with PBS and incubated in 0.5% (*w/v*) crystal violet (SERVA Electrophoresis, Heidelberg, Germany) in 20% (*v/v*) methanol (Merck, Darmstadt, Germany) for 30 min at RT followed by rinsing with aqua dest. Colonies were defined as circular arrangements of cells of more than 50 stained cells, indicating that one viable cell as colony forming unit gave rise to a colony through replication [13,62].

### 4.5. Characterization of Mesenchymal Stromal Cells

Following the International Society for Cellular Therapy’s (ISCT) minimal criteria to define mesenchymal stromal cells (MSCs), plastic adherence, as well as appropriate surface marker expression and trilineage differentiation, was chosen for MSC characterization [63,64,65,66].

### 4.6. Flow Cytometry

After the 3rd passage (cultivation period of app. 3 weeks), cells were detached, resuspended and counted. Aliquots of approximately 1 × 10^5^ cells were incubated with antibodies against CD31 (APC-eFluor 780, clone WM59; eBioscience/Thermo Fisher Scientific, Carlsbad, CA, USA), CD34 Class III (FITC, clone: 581, BioLegend, Fell, Germany), CD45 (V500, clone: HI30, Becton Dickinson BD Bioscience, Heidelberg, Germany), CD 29 (PE, clone MAR4, Becton Dickinson), CD73 (PerCP-eFlour-710, clone: AD2, eBioscience), CD90 (Brilliant Violet 421, clone: 5E10, BioLegend) and CD105 (PE-Cy7, clone: 43A3, BioLegend) for 30 min on ice, as described before [22]. Isotype controls at the same concentration as the specific antibodies were used to determine nonspecific signals. Flow cytometry was performed with a FACSCanto II flow cytometer (BD Bioscience) and Diva Software v6.0 (BD Bioscience).

### 4.7. Differentiation of the Mesenchymal Stem Cells

Mesenchymal multipotency was approved by applying typical in vitro stimulation protocols with the respective media, followed by representative cytochemical staining. In all groups, unstimulated cells served as controls [67].

Osteogenic differentiation: 2 × 10^4^ cells were cultivated in a 12-well plate in osteogenic medium (StemPro™ Osteogenesis Differentiation Kit, Thermo Fisher Scientific, Dreieich, Germany). After 21 days, cells were fixed in 4% (*v/v*) formalin (Merck Millipore, Darmstadt, Germany), rinsed in PBS, and the mineralization of the extracellular matrix of the osteoblasts was stained with 2% (*w/v*) alizarin red (LifeLine Cell Technology, Oceanside, CA, USA).Chondrogenic differentiation: Cells were concentrated to 1.6 × 10^7^ cells/mL and an aliquot of 8 × 10^4^ cells per 5 µL was inserted per well of a 96-well round bottom plate. Cells were cultured in chondrogenic media (StemPro™ Chondrogenesis Differentation Kit) in 96-well plates. After 21 days, the cell pellet was fixed in 4% (*v/v*) formalin, rinsed with PBS and the glycosaminoglycans of the chondrocytes stained with 1% (*w/v*) alcian blue in 0.1 N HCl (Roth, Karlsruhe, Germany) for 30 min at RT. Differentiation was performed with 0.1 N HCl.Adipogenic differentiation: 4 × 10^4^ cells were cultivated in a 12-well plate in adipogenic medium (StemPro™ Adipogenesis Differentation Kit). After 21 days, cells were fixed in 10% (*v/v*) formalin (Applichem, Darmstadt, Germany), washed in aqua dest and in 60% (*v/v*) isopropanol (Applichem) for 5 min. The oil vacuoles of the adipocytes were detected by staining with 0.18% (*w/v*) oil-red-O (Sigma) in 60% (*v/v*) isopropanol.

### 4.8. Human Cytokine Array Analysis

The mixture of BSM and cell–tissue composite was tested for its relative expression levels of 102 human soluble proteins (cytokines, chemokines and growth factors and other extracellular signaling molecules) using the human cytokine array (ARY022, R&D Systems, Abingdon, UK) (a complete list of the 102 soluble proteins is available in the Appendix A).

The samples were thawed and analyzed according to the manufacturer’s instructions. In brief, the microarray was blocked with blocking buffer for 60 min at RT; subsequently, each well was overlaid with 100 mL of diluted sample (1:200 in assay diluent) and left at 4 °C for 12 h. The microarray slide was washed with washing buffer and incubated with biotinylated secondary antibody for 60 min at RT. After another wash, horseradish-peroxidase (HRP) conjugated streptavidin was added for 30 min at RT. As substrate, luminol was used and the signal was detected after 0.5, 1 or 5 min using an X-ray film in an autoradiography cassette. After film processing, the film was documented.

### 4.9. Statistics

Statistical analysis was performed using Graph Pad Prism software V8 (GraphPad Prism Software, Inc. San Diego, CA). Continuous variables (patients’ age, sample weight, MNC number, generation time) are presented as mean ± standard deviation and categorical variables (gender) as frequency and percentage. Ordinal parameters (CFU number) are expressed as median with the interquartile range (25th–75th percentile). Analysis of normal distribution of each continuous variable was performed by the Kolmogorov–Smirnov test before further statistical testing. Accordingly, the Mann–Whitney U test was used for comparison of nonparametric values (sample weight, MNC number, order of BSM used) and the unpaired *t*-test for parametric values (number of CFU, generation time, ratio MSC per million MNCs) between the two study groups. Differences were considered significant at *p* < 0.05.

## 5. Conclusions

We conclude that, during orthopedic surgery, tissue with relevant amounts of MSCs and growth factors can be concentrated by an advanced surgical vacuum handle filled with bone substitute material. This new potent source of autologous MSC and growth factors could be identified. The innovation is that well-established osteoconductive materials can be enriched with osteoinductive cytokines and MSC-containing tissue without additional equipment. We believe that this device has potential for clinical application as a medicinal product. Here, clinical tests can easily be achieved using the BoneFlo system as a container for the investigation of different biomaterials.

## Figures and Tables

**Figure 1 ijms-21-06393-f001:**
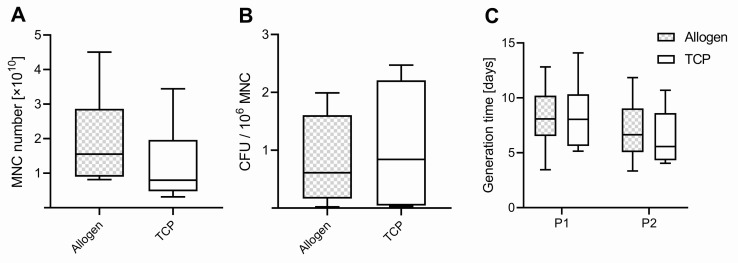
(**A**) Number of mononuclear cells (MNC), (**B**) colony-forming units (CFU) per million MNCs, and (**C**) generation time of passage 1 (P1) and 2 (P2). Boxplots indicate the median within the 25–75% percentile.

**Figure 2 ijms-21-06393-f002:**
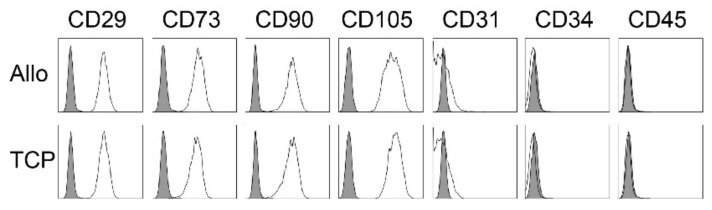
Representative flow cytometry analyses for MSC from cell-tissue composites (CTC) from allogenic (Allo) and synthetic (TCP) bone substitute material for one patient. Data are shown as a histogram overlay: isotype control (grey) and specific cell surface markers (white). Cells were labeled with antibodies against CD29, CD73, CD90, CD105, CD31, CD34 and CD45.

**Figure 3 ijms-21-06393-f003:**
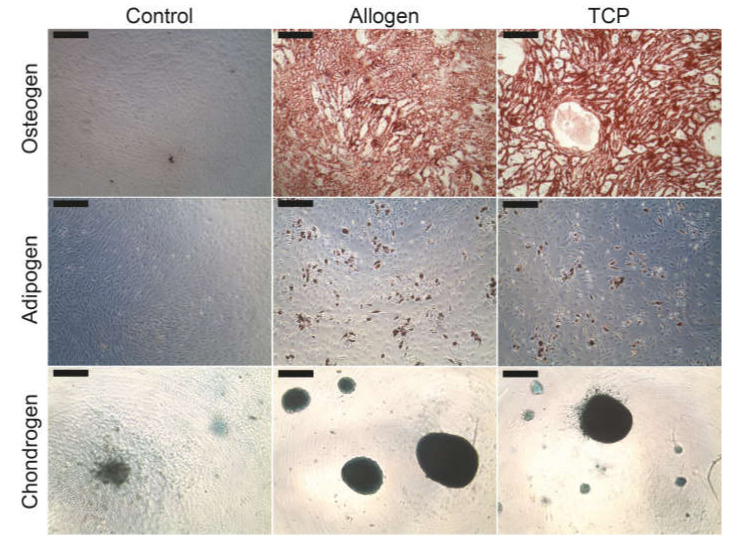
Differentiation of the MSC into osteoblasts, adipoblasts and chondroblasts. The scale bar represents 200 µm.

**Figure 4 ijms-21-06393-f004:**
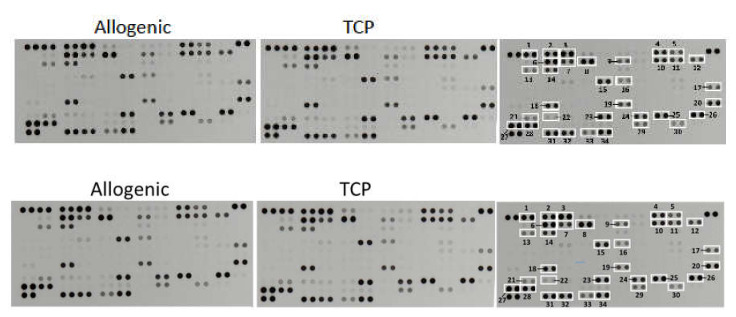
Representative X-ray results of the cytokine array from the cell–tissue composites (Allo vs. TCP) of one patient. The right scheme displays the respective coordinate reference number # for analyte identification. Unframed spots are assay-specific reference spots.

**Figure 5 ijms-21-06393-f005:**
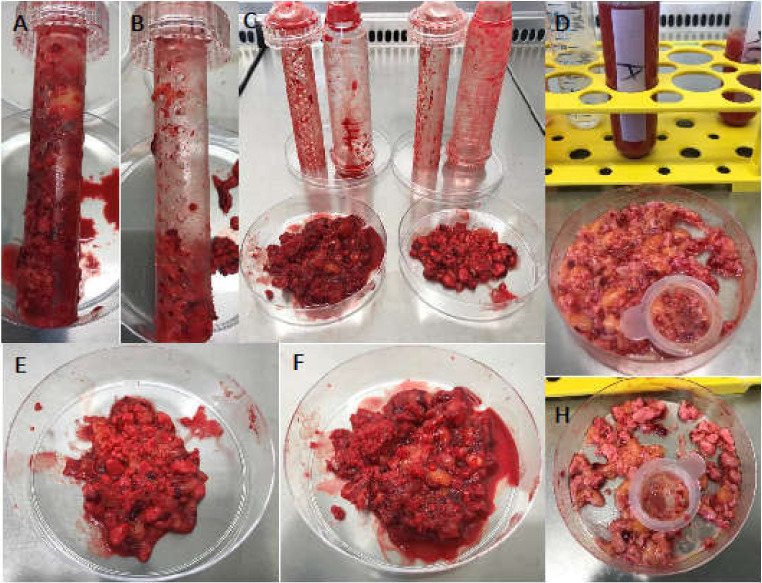
Content of the surgical vacuum suction handle after application in arthroplasty surgeries. Surgical vacuum suction device’s inner part filled with allogenic (Allo) or synthetic (TCP) bone substitute material (BSM) (**A**,**F**) (**B**,**E**). The polystyrene container was opened under the laminar flow bench. The BSM and retained cell-tissue composite (CTC) were placed in petri dishes (**C**) and washed with phosphate-buffered saline (PBS) and filtered. (**D**) Residual allogenic BSM (**D**), residual TCP (**H**).

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
