# Peer review of "Ceramic Scaffolds in a Vacuum Suction Handle for Intraoperative Stromal Cell Enrichment"

_ijms, 2020, doi:10.3390/ijms21176393_

Round 1
Reviewer 1 Report
The study shows how a vacuum suction handle filled with a bone substitute material (BSM) can act as a trap for cell-tissue-composites (CTC), containing pro-regenerative cytokines and mononuclear cells, among which a certain proportion are mesenchymal stem cells, with proliferative and tri-lineage differentiation potential. Two different BSM have been used, an organic one (cancellous bone allograft), and an inorganic one (beta-TCP), with no significant differences found between them in terms of number of mononuclear cells, CFU or cytokine profile yield.
The authors consider, and this reviewer agrees with the idea, that the so-collected CTC have a potential clinical application in orthopedics as promoters of bone repair.
Given the limitations that EU normative imposes on the use of stem cells, there is an interest, particularly among traumatologists, about in-operation strategies that allow the use of autologous MSC. The one here proposed has been around for some time but, to my knowledge, this is the first report of a strategy involving a suction handle filled with BSM, as a design to trap CTC. The design is simple and ingenious, and the authors show it traps CTC containing MSC and cytokines.
Said this, evidence is not provided about the real effectiveness of the procedure. Without an in vivo assay, or some other type of functional assay, this article does not add much to their previous paper:
Henze, K., Herten, M., Haversath, M. et al. Surgical vacuum filter-derived stromal cells are superior in proliferation to human bone marrow aspirate. Stem Cell Res Ther 10, 338 (2019). https://doi.org/10.1186/s13287-019-1461-0
Minor comments:
Page 6. Lines 194-196
It cannot be guaranteed the device provides celllular adherence because no experimental procedure has ben used to actually show cells adhered to the biomaterial (SEM, for example). Cells may perfectly be located in CTC trapped among the biomaterial chips.
Author Response
Reviewer 1
The study shows how a vacuum suction handle filled with a bone substitute material (BSM) can act as a trap for cell-tissue-composites (CTC), containing pro-regenerative cytokines and mononuclear cells, among which a certain proportion are mesenchymal stem cells, with proliferative and tri-lineage differentiation potential. Two different BSM have been used, an organic one (cancellous bone allograft), and an inorganic one (beta-TCP), with no significant differences found between them in terms of number of mononuclear cells, CFU or cytokine profile yield.
The authors consider, and this reviewer agrees with the idea, that the so-collected CTC have a potential clinical application in orthopedics as promoters of bone repair.
Given the limitations that EU normative imposes on the use of stem cells, there is an interest, particularly among traumatologists, about in-operation strategies that allow the use of autologous MSC. The one here proposed has been around for some time but, to my knowledge, this is the first report of a strategy involving a suction handle filled with BSM, as a design to trap CTC. The design is simple and ingenious, and the authors show it traps CTC containing MSC and cytokines.
Said this, evidence is not provided about the real effectiveness of the procedure. Without an in vivo assay, or some other type of functional assay, this article does not add much to their previous paper:
Henze, K., Herten, M., Haversath, M. et al. Surgical vacuum filter-derived stromal cells are superior in proliferation to human bone marrow aspirate. Stem Cell Res Ther 10, 338 (2019). https://doi.org/10.1186/s13287-019-1461-0
Thank you for taking the time and effort to work through the manuscript and the feedback for improving its quality.
We thank the reviewer for this critical point. In our opinion, the actual manuscript investigated the practical implementation of a conceptual progression of the vacuum suction handle.
In our previous paper (Henze et al.), we could show that the cell-tissue-composites (CTC) contained MSC with osteogenic regeneration potential. In our actual manuscript, we combined the CTC with bone substitute material and investigated the character of the trapped cells and the micro milieu in respect to the cytokine pattern. As a consequence we are planning an in vivo experiment, in which bone substitute material with autologous CTC will be used for bone defect repair in an animal model.
Minor comments:
Page 6. Lines 194-196
It cannot be guaranteed the device provides cellular adherence because no experimental procedure has been used to actually show cells adhered to the biomaterial (SEM, for example). Cells may perfectly be located in CTC trapped among the biomaterial chips.
Thank you very much, this is correct and a good point for discussion. Indeed, we can not distinguish between trapping of the cells among the biomaterial chips or cellular adherence to the bone replacement material. We added this to the discussion section within the limitations of the study (page 8 line 296-298).
Reviewer 2 Report
Excellent literature and new novel ideas within the ortho space with arthroplasty. The changes within the arthroplasty with the stomal cells are interesting and effects are critical to understand the long term affects
Author Response
Thank you for taking the time and effort to work through the manuscript and the feedback for improving its quality.
Reviewer 3 Report
General comments
The submitted manuscript reports on the investigation of the osteopromotive potential of cell-tissue-composites combined with bone substitute filler material (i.e. tricalcium phosphate and cancellous allograft) during surgical suction, considering two different surgeries, namely total hip and knee replacement surgeries.
The present work does not seem an Article/ Full Research Paper, but sounds more as a Technical Paper. There are many criticisms and weaknesses, such as the limited number of studied cases (10), the involvement of different tissue compartments during surgery, and thus the impossibility to identify the MSCs and growth factors source, the investigation of tissue samples from two different surgical procedures (total hip or knee replacement), the use of PS suction device, as evidenced by the same authors. The data are not deeply described, and the discussion is too generic and not specifically correlated to the results acquired by the authors. Thus, both Results and Discussion sections have to be improved and expanded, accordingly.
Finally, a deep English grammar and language revision is strongly suggested.
Specific comments and remarks are listed below point by point.
Keywords
Among the chosen keywords (i.e. bone defect; regeneration; biomaterial; ceramic filter; bone remodeling; surgical suction), ‘biomaterial’ is too generic. Moreover, further specific keywords, such as the used ceramic fillers, the kind of surgeries, and so on, could be added.
It is also suggested to report the keywords in a more logical order (i.e. materials, processing, characterisations, properties, applications).
- Introduction
- The Introduction section is well organised and conceived, even if the authors have to evidence the originality of their work with respect to the literature.
- All the following period “Therefore, a large amount of boths, natural (allo-/xenografts) and synthetic (e. g. tricalcium phosphate [TCP], hydroxyapatite [HA], calcium sulfate, calcium carbonate, composites) bone substitutes have been applied in orthopaedic and trauma surgery for a long time. In contrast to autologous bone, these materials are not vital and mainly act as an osteoconductive scaffold for local tissue ingrowth. Other bone substitutes pledge an additional osteoinductive potential with favorable effects on proliferation and differentiation of local cells. Osteoinductive materials are potent to induce new bone formation even in a non-osseous environment by recruiting undifferentiated and pluripotent progenitor cells followed by differentiation into bone-forming cells” has to be supported with proper literature references. Also bioactive glasses are used for these applications and they have to be cited among the used synthetic materials.
- Similarly, the authors have to support the following statement “However, the harvesting of these cells intraoperatively is time-consuming, there are sterility concerns as the aspirate may leave the sterile field and there is also a risk to change the biology of the cells depending on following processing of the tissue.” with proper references.
- The period “In orthopaedics and trauma surgery, hard tissue opening or damage is frequent and associated with a relevant amount of bone marrow leaking out of the bone. Furthermore, the surgical approach and/or trauma includes also soft-tissue damage and hematoma. As a consequence, not only bone fragments, but also small pieces of muscle, fat, connective tissue and vessels are liberated from the surrounding tissue. Some of these are removed by surgical suction, some are left in the situs” has to be corroborated with appropriate references.
- Finally, also the period “During surgery, most surgeons use vacuum suction devices to get optimal view over surgical site. Moreover, these systems can be combined with cell safer devices if relevant blood loss is assumed (autologous blood transfusion). Especially during total joint replacement or other types of operations demanding a more extensive surgical approach, relevant amounts of the above-mentioned mixture of cytokines, growth factors and cells and tissue (cell-tissue composites, CTC) will get lost by vacuum suction.” needs suitable references.
- The Authors have to better highlight the originality and added value of their work not only with respect to their previous paper, but also with respect to the other previous literature reports about the same topic.
- It is recommendable to indicate at the end of the Introduction section the main employed characterisation techniques in order to achieve the indicated purpose.
- Results
The paragraphs 2.1, 2.2 and 2.3 should be combined in a unique one in order to make the description and discussion more synergic. Moreover, all the results have to be more deeply described.
Similarly, the paragraphs 2.4 and 2.4 should be combined.
2.1. Yield of mononuclear cells (MNC)
- For the total amount of harvested mononuclear cells for Allo (1.94 ± 1.32 x 10^10) and TCP (1.26 ±1.03 x 10^10), the standard deviations are too high. Please justify these results.
2.2. Stemness character and MSC yield
- As evidenced for the total amount of harvested mononuclear cells, also the reported CFU values (Fig. 1B) are characterised by too high standard deviations (for TCP it is equivalent to the average value).
- Concerning the ratio of potential MSC per MNC harvested, for TCP the standard deviation was higher than the average value.
2.5. Differentiation potential: differentiation into osteoblasts, chondroblasts and adipoblasts
- In Fig. 3 the scale bar is not visible. Please replace it.
- This paragraph has to be combined with the previous one (2.4) and has to be better described.
- Discussion
- The discussion section has to be reorganised. In many points the discussion seems not correlated to the results obtained and acquired by the authors but very general, reporting the description of data from previous papers, without a comparison with their work.
- The following conclusion “However, the surgical vacuum suction device guaranties wetting of the bone substitute as well as cellular adherence without any additional invasive procedure and without further tissue or cell processing” has to be better discussed and supported with literature references.
- The authors stated that “It is unclear if a longer application of the vacuum is able to increase the amount of MNC in the filter biomaterial”..They should investigate this aspect, carrying out experiments for longer times.
- The authors affirmed that “In our study, the suction device was made of polystyrene (OP-Flex™ FilterFlow™, ConvaTec, Deeside, UK), which is not qualified for in orthopaedic vivo-application. .. Due to patient´s safety issues the authors recommend a more biocompatible and inert material (e. g. polymethylenacrylate, PMMA) if clinical application is intended”. Thus, why did the use PS? It has to be clearly explained and justified.
- The following consideration “Besides the application site (including cellular components), the type of vacuum (level, duration), the suction handle´s material, the type of bone substitute might have an impact for in vitro results.” has to be supported with suitable references.
- The authors have to corroborate the following statement “However, the application of new coating strategies and the development of composites (e.g. polycaprolactone-tricalcium phosphate [PCL-TCP]) have a potential to improve currently available materials.” with proper references.
- Appropriate references for the ex vivo and in vivo approaches have to be added.
- Materials and methods
4.1. Patients
Concerning the elective total hip (n=6) or knee (n=4) replacement, the authors have to specify how many men and how many women for the hip and knee replacement.
4.3. Isolation and cultivation of the cells
- In the following sentence “The mixture of BSM and cell tissue-composite (Fig. 5E, Fig. 5F) was incubated with streptokinase (3 KU; S3134, Sigma Aldrich, Steinheim, Germany) in 10 ml PBS at RT for 15 min”, the weight of the used BSM/cell tissue-composite mixture has to be specified.
- Conclusions
The Conclusions section has to be improved. It is too short. The authors should highlight the main results.
Author Response
Reviewer 3
General comments
The submitted manuscript reports on the investigation of the osteopromotive potential of cell-tissue-composites combined with bone substitute filler material (i.e. tricalcium phosphate and cancellous allograft) during surgical suction, considering two different surgeries, namely total hip and knee replacement surgeries.
The present work does not seem an Article/ Full Research Paper, but sounds more as a Technical Paper. There are many criticisms and weaknesses, such as the limited number of studied cases (10), the involvement of different tissue compartments during surgery, and thus the impossibility to identify the MSCs and growth factors source, the investigation of tissue samples from two different surgical procedures (total hip or knee replacement), the use of PS suction device, as evidenced by the same authors. The data are not deeply described, and the discussion is too generic and not specifically correlated to the results acquired by the authors. Thus, both Results and Discussion sections have to be improved and expanded, accordingly.
Finally, a deep English grammar and language revision is strongly suggested.
Specific comments and remarks are listed below point by point.
Keywords
Among the chosen keywords (i.e. bone defect; regeneration; biomaterial; ceramic filter; bone remodeling; surgical suction), ‘biomaterial’ is too generic. Moreover, further specific keywords, such as the used ceramic fillers, the kind of surgeries, and so on, could be added.
It is also suggested to report the keywords in a more logical order (i.e. materials, processing, characterisations, properties, applications).
Thank you for taking the time and effort to work through the manuscript and the feedback for improving its quality.
Thank you for this suggestion. We added “ß-TCP, Total Joint Arthroplasty, Allograft” and arranged the keywords in a new order:” bone defect; bone remodelling; surgical suction; ceramic filter; ß-TCP; allograft; total joint arthroplasty”. (page 1, line 36-37).
- Introduction
- The Introduction section is well organised and conceived, even if the authors have to evidence the originality of their work with respect to the literature.
Thank you for this suggestion. We added in the introduction (page 2, line 93-96): “To our knowledge, it is the first device which is able to enrich autologous tissue onto a bone substitute without any additional surgical approach or other time consuming interoperative procedures. Moreover, the cell/tissue-TCP/allograft composite does not required further processing outside the surgical site.”
- All the following period “Therefore, a large amount of boths, natural (allo-/xenografts) and synthetic (e. g. tricalcium phosphate [TCP], hydroxyapatite [HA], calcium sulfate, calcium carbonate, composites) bone substitutes have been applied in orthopaedic and trauma surgery for a long time.x1 In contrast to autologous bone, these materials are not vital and mainly act as an osteoconductive scaffold for local tissue ingrowth. Other bone substitutes pledge an additional osteoinductive potential with favorable effects on proliferation and differentiation of local cells. Osteoinductive materials are potent to induce new bone formation even in a non-osseous environment by recruiting undifferentiated and pluripotent progenitor cells followed by differentiation into bone-forming cells” has to be supported with proper literature references. Also bioactive glasses are used for these applications and they have to be cited among the used synthetic materials.
Thank you for this point. We added the appropriate references [6-9]. Additionally we included that: “In the last few years, mesoporous bioactive glasses (MBGs) are becoming increasingly popular as bone substitute materials [10].” (Page 2, line 51-53).
- Similarly, the authors have to support the following statement “However, the harvesting of these cells intraoperatively is time-consuming, there are sterility concerns as the aspirate may leave the sterile field and there is also a risk to change the biology of the cells depending on following processing of the tissue.” with proper references.
Thank you for this point. We added the appropriate references [16-20] (page 2, line 67).
- The period “In orthopaedics and trauma surgery, hard tissue opening or damage is frequent and associated with a relevant amount of bone marrow leaking out of the bone. Furthermore, the surgical approach and/or trauma includes also soft-tissue damage and hematoma. As a consequence, not only bone fragments, but also small pieces of muscle, fat, connective tissue and vessels are liberated from the surrounding tissue. Some of these are removed by surgical suction, some are left in the situs” has to be corroborated with appropriate references.
Thank you for this point. We added an appropriate reference [22] (page 2, line 78).
- Finally, also the period “During surgery, most surgeons use vacuum suction devices to get optimal view over surgical site. Moreover, these systems can be combined with cell safer devices if relevant blood loss is assumed (autologous blood transfusion). Especially during total joint replacement or other types of operations demanding a more extensive surgical approach, relevant amounts of the above-mentioned mixture of cytokines, growth factors and cells and tissue (cell-tissue composites, CTC) will get lost by vacuum suction.” needs suitable references.
Thank you for this point. We added the appropriate references [24,25] (page 2 line 90).
- The Authors have to better highlight the originality and added value of their work not only with respect to their previous paper, but also with respect to the other previous literature reports about the same topic.
Thank you for this point. We added that: (page 2, line 94 – 97):”To our knowledge, it is the first device which is able to enrich autologous tissue onto a bone substitute without any additional surgical approach or other time consuming interoperative procedures. Moreover, the cell/tissue-TCP/allograft composite does not required further processing outside the surgical site.
- It is recommendable to indicate at the end of the Introduction section the main employed characterisation techniques in order to achieve the indicated purpose.
Thank you very much for this suggestion. We added (page 3 line 98-110):”There are many research groups who working on solutions for bone regeneration onto biomaterials [26]. In contrast to other investigators, we intend a simple, time-saving and patient-friendly solution. Compared to other systems such the Reamer Irrigator Aspirator [27] or bone aspiration techniques with or without density gradient cell centrifugation (e.g. Harvest System [28], ASPIRE™ Bone Marrow Harvesting System [29] no additional surgical approach and device material is required. Other groups prefer PRP or other products from the peripheral blood to treat bone defects [30,31].
Although the content of patelet lysates are highly promising to regenerative approaches in translational orthopaedics [32] and these systems are also non-invasive (i. v. punction), they leave the surgical site and do not penetrate porous bone substitutes as a vacuum system can perform. At least, most systems designed for bone regeneration are a pre-clinical stage [33] A simple solution such as a vacuum suction handle combined with a bone substitute might enter the market as a medicinal product much easier by an extension of an established certificate (suction handle only).”
- Results
The paragraphs 2.1, 2.2 and 2.3 should be combined in a unique one in order to make the description and discussion more synergic. Moreover, all the results have to be more deeply described.
Similarly, the paragraphs 2.4 and 2.4 should be combined.
We combined the former sections 2.1 – 2.3 to “2.1 Comparison of the MNC yield, stemness character and proliferation potential” and the former paragraphs 2.4 and 2.5 with “Cell characterization by flow cytometry” and “Differentiation potential: differentiation into osteoblasts, chondroblasts and adipoblasts” to the new paragraph 2.2. “Differentiation potential”. We added some more results and hope that we answered your suggestion to your satisfaction. (page 3, line 113 – page 4 line 158).
2.1. Yield of mononuclear cells (MNC)
- For the total amount of harvested mononuclear cells for Allo (1.94 ± 1.32 x 10^10) and TCP (1.26 ±1.03 x 10^10), the standard deviations are too high. Please justify these results.
The weight of the BSM after the vacuum suction varied for both BSM. Partially, in some cases a few BSM granules were spilled within the filling process, also the weight of the resulting bone substitute material with CTC material varied from patient to patient, especially in obese patients. We added this in the discussion (page 9, line 329 – 332).
We correlated the sample weight with the amount of harvested MNC and reached a significant correlation of 0.606 (Pearson) with a p-value of 0.006 and added this in the result section (page 3 line 121 – 122).
If desired, we can add the data of the BSM type, BSM + CTC weight after suction and the number of MNC per BSM for each patient.
2.2. Stemness character and MSC yield
- As evidenced for the total amount of harvested mononuclear cells, also the reported CFU values (Fig. 1B) are characterised by too high standard deviations (for TCP it is equivalent to the average value).
- Concerning the ratio of potential MSC per MNC harvested, for TCP the standard deviation was higher than the average value.
We are aware, that the parameters (amount of harvested MNC, CFU/Mill CFU and the MSC per 1000 MNC) show standard variations higher than the mean values. We have added the range in order to show this variation.
2.5. Differentiation potential: differentiation into osteoblasts, chondroblasts and adipoblasts
- In Fig. 3 the scale bar is not visible. Please replace it.
Thank you for pointing out this imprecision. The scale bar was replaced by a better visible one.
- This paragraph has to be combined with the previous one (2.4) and has to be better described.
The former paragraphs 2.4 and 2.5 were combined.
Discussion
- The discussion section has to be reorganised. In many points the discussion seems not correlated to the results obtained and acquired by the authors but very general, reporting the description of data from previous papers, without a comparison with their work.
The authors thank the reviewer for this helpful and critical hint. The major problem is that there are no data of comparable systems available in the literature. In order to follow the internal logic of the study design (material & methods, results) we discussed at first the question of critical MSCs numbers potent for bone regeneration followed by other aspects. We are happy to present appropriate subheadings to the reader which might facilitate following the thread.
- Influence of the MSC number on bone regeneration
- Influence of the plastic container and vacuum
- Impact of the biomaterial and regulatory aspects
- Conditioning systems, composites and growth factors
We hope that this will meet the requirements of the reviewer´s intension.
The following conclusion “However, the surgical vacuum suction device guaranties wetting of the bone substitute as well as cellular adherence without any additional invasive procedure and without further tissue or cell processing” has to be better discussed and supported with literature references.
Thank you very much for this point. We added the corresponding references [17, 37] (page7, line 224).
- The authors stated that “It is unclear if a longer application of the vacuum is able to increase the amount of MNC in the filter biomaterial”.They should investigate this aspect, carrying out experiments for longer times.
Thank you for your suggestion. We should investigate the influence of the application time on the yield of MNC in the bone substitute material in further experiments.
- The authors affirmed that “In our study, the suction device was made of polystyrene (OP-Flex™ FilterFlow™, ConvaTec, Deeside, UK), which is not qualified for in orthopaedic vivo-application. Due to patient´s safety issues the authors recommend a more biocompatible and inert material (e. g. polymethylenacrylate, PMMA) if clinical application is intended”. Thus, why did the use PS? It has to be clearly explained and justified.
Thank you for your comment. We did not use a suction device which is made of PMMA because there is none available. Polysterene is one of the most applied materials for suction devices in surgery.
- The following consideration “Besides the application side (including cellular components), the type of vacuum (level, duration), the suction handle´s material, the type of bone substitute might have an impact for in vitro results.” has to be supported with suitable references.
Thank you very much for this point. We added the corresponding references: “Besides the application site (including cellular components) [37-39], the type of vacuum (level, duration) [17, 37], the suction handle´s material [38,39], the type of bone substitute [40,41] might have an impact for in vitro results.” (page7, line 235-237).
- The authors have to corroborate the following statement “However, the application of new coating strategies and the development of composites (e.g. polycaprolactone-tricalcium phosphate [PCL-TCP]) have a potential to improve currently available materials.” with proper references.
Thank you very much for this point. We added the corresponding references [46,47] (page 7, line 251).
Appropriate references for the ex vivo and in vivo approaches have to be added.
Thank you very much for this point. We added the corresponding references [17,37] (page 8, line 259).
Materials and methods
4.1. Patients
Concerning the elective total hip (n=6) or knee (n=4) replacement, the authors have to specify how many men and how many women for the hip and knee replacement.
From 6 patients receiving total hip arthroplasty there were four women and two men. Among the four patients who received total knee arthroplasty there were three men and 1 woman.
4.3. Isolation and cultivation of the cells
- In the following sentence “The mixture of BSM and cell tissue-composite (Fig. 5E, Fig. 5F) was incubated with streptokinase (3 KU; S3134, Sigma Aldrich, Steinheim, Germany) in 10 ml PBS at RT for 15 min”, the weight of the used BSM/cell tissue-composite mixture has to be specified.
The volume of 10 ml PBS and the amount of the streptokinase (3KU) was the same for all BSM/cell tissue-composite mixture. Since the sample weight varied, the quotient of g sample/K units streptokinase also varied with 4.6 ± 1.87 g tissue/KU streptokinase for Allo and with 5.3 ± 1.50 g tissue/KU streptokinase for TCP.
Conclusions
The Conclusions section has to be improved. It is too short. The authors should highlight the main results.
Thank you for this point. We added that:“The innovation is that well-established osteoconductive materials can be enriched with osteoinductive cytokines and MSC-containing tissues without additional equipment. We belief that this device has potential for clinical application as a medicinal product. Here clinical test can easy achieved using the BoneFlo system as a container for the investigation of different biomaterials.” (page12, line 472-476.)
Round 2
Reviewer 3 Report
General comments
The Authors have replied to all the Reviewers’ questions and remarks. A minor revision is requested before considering this paper for publication in IJMS, as reported below.
- Materials and methods
4.1. Patients
- Even if the authors replied to the following request “Concerning the elective total hip (n=6) or knee (n=4) replacement, the authors have to specify how many men and how many women for the hip and knee replacement”, the related information were not reported within the manuscript text.
Author Response
- Materials and methods
4.1. Patients
- Even if the authors replied to the following request “Concerning the elective total hip (n=6) or knee (n=4) replacement, the authors have to specify how many men and how many women for the hip and knee replacement”, the related information were not reported within the manuscript text
Thank you for revising the manuscript and the feedback for improving its quality.
We have now specified the gender distribution also within the manuscript (page9, line 342-343):”Following a prospective design, the patient cohort consisted of 10 patients (5 females, 5 males, mean age of 66.8 ± 9.1 years) with advanced osteoarthritis qualified for elective total hip (n=6; four females, two males) or knee (n=4; one females, three males) replacement.”